# EEG Monitoring Is Feasible and Reliable during Simultaneous Transcutaneous Electrical Spinal Cord Stimulation

**DOI:** 10.3390/s21196593

**Published:** 2021-10-02

**Authors:** Ciarán McGeady, Aleksandra Vučković, Yong-Ping Zheng, Monzurul Alam

**Affiliations:** 1Centre for Rehabilitation Engineering, University of Glasgow, Glasgow G12 8QQ, UK; c.mcgeady.1@research.gla.ac.uk; 2Department of Biomedical Engineering, The Hong Kong Polytechnic University, Hung Hom, Kowloon, Hong Kong; yongping.zheng@polyu.edu.hk

**Keywords:** transcutaneous spinal cord stimulation, electroencephalography, artifact removal, brain–computer interface, BCI, rehabilitation

## Abstract

Transcutaneous electrical spinal cord stimulation (tSCS) is a non-invasive neuromodulatory technique that has in recent years been linked to improved volitional limb control in spinal-cord injured individuals. Although the technique is growing in popularity there is still uncertainty regarding the neural mechanisms underpinning sensory and motor recovery. Brain monitoring techniques such as electroencephalography (EEG) may provide further insights to the changes in coritcospinal excitability that have already been demonstrated using other techniques. It is unknown, however, whether intelligible EEG can be extracted while tSCS is being applied, owing to substantial high-amplitude artifacts associated with stimulation-based therapies. Here, for the first time, we characterise the artifacts that manifest in EEG when recorded simultaneously with tSCS. We recorded multi-channel EEG from 21 healthy volunteers as they took part in a resting state and movement task across two sessions: One with tSCS delivered to the cervical region of the neck, and one without tSCS. An offline analysis in the time and frequency domain showed that tSCS manifested as narrow, high-amplitude peaks with a spectral density contained at the stimulation frequency. We quantified the altered signals with descriptive statistics—kurtosis, root-mean-square, complexity, and zero crossings—and applied artifact-suppression techniques—superposition of moving averages, adaptive, median, and notch filtering—to explore whether the effects of tSCS could be suppressed. We found that the superposition of moving averages filter was the most successful technique at returning contaminated EEG to levels statistically similar to that of normal EEG. In the frequency domain, however, notch filtering was more effective at reducing the spectral power contribution of stimulation from frontal and central electrodes. An adaptive filter was more appropriate for channels closer to the stimulation site. Lastly, we found that tSCS posed no detriment the binary classification of upper-limb movements from sensorimotor rhythms, and that adaptive filtering resulted in poorer classification performance. Overall, we showed that, depending on the analysis, EEG monitoring during transcutaneous electrical spinal cord stimulation is feasible. This study supports future investigations using EEG to study the activity of the sensorimotor cortex during tSCS, and potentially paves the way to brain–computer interfaces operating in the presence of spinal stimulation.

## 1. Introduction

Transcutaneous electrical spinal cord stimulation (tSCS) is a non-invasive neuromodulatory technique that has shown promise in recent years in promoting the motor recovery of spinal-cord injury patients [1,2,3]. The technique uses a surface electrode positioned over the site of spinal injury to deliver high-frequency currents, and has been associated with functional improvements in the upper limbs, the trunk [4], and the lower limbs [1], often when combined with physical practice. It has been postulated that tSCS elevates the motor threshold of dorsal root motoneurons, making volitional control easier through residual descending pathways [5]. The precise mechanisms underpinning recovery, however, are not fully understood. Recent studies have used various techniques to measure changes in corticospinal excitability, one even reporting changes to cortical excitability after tSCS [6]. Going forward it will be crucial to explore numerous research avenues, employing a range of techniques, such as electroencephalography (EEG), in order to establish the precise mechanisms of recovery.

As with other stimulation-based techniques—functional electrical stimulation [7], transcranial direct current stimulation [8], transcranial alternating current stimulation (tACS) [9], deep brain stimulation [10], for example—the introduction of EEG into an experiment may present a significant challenge when it comes to interpretation, owing to substantial stimulation artifacts in the recorded signal. Stimulation is often applied at intensities far exceeding the amplitudes associated with EEG. A pitfall may present itself in the frequency domain if the stimulation frequency overlaps with a frequency range of interest, indeed cervical tSCS is often delivered at 30 Hz, within EEG’s sensorimotor spectrum (7–40 Hz) [11]. Many tACS studies have overcome this conflict by limiting their EEG analysis to before and after stimulation. This removes the artifact problem but deprives the study of access to brain activity during stimulation. Recently, EEG during continuous tACS was monitored and artifacts were removed with artifact-suppression techniques [12], to an extent allowing the analysis of brain rhythms during stimulation. Transcranial alternating current stimulation (tACS) has as similar periodic waveform to tSCS. Artifact-suppression techniques developed for tACS are a good starting point for examining tSCS and EEG.

Whether tSCS complicates the extraction of neural information has not, to the best of our knowledge, been reported. This study aims to cast light on the way tSCS manifests on EEG during simultaneous acquisition, to quantify its effects and determine if artifact-suppression techniques can be used to minimise contamination. To address these questions we gathered an EEG dataset from healthy volunteers while stimulation was delivered transcutaneously to the cervical region on the posterior side of the neck. The location of stimulation and stimulation parameters—carrier frequency, burst frequency, pulse width, etc.—were chosen to reflect parameters typical of the current state-of-the-art in upper-limb rehabilitation using tSCS [2,3]. We performed an offline analysis to illustrate how tSCS manifested in the time and frequency domain and considered the impact of EEG electrode location and stimulation intensity on artifact prominence. Our hypothesis was that, like other electrical stimulation techniques, tSCS would present in EEG as narrow, high-amplitude peaks and that artifact-suppression techniques could reduce the impact of stimulation. Overall, our results implied that extracting physiologically meaningful EEG during tSCS is possible, and paves the way to future research aimed at uncovering the sensorimotor neural mechanisms behind tSCS-based therapy.

## 2. Materials and Methods

### 2.1. Participants

Twenty-one healthy volunteers (7 females, 14 males; 28 ± 5 years old) participated in this study. Exclusion criteria included previous neurological symptoms of the nervous or musculoskeletal systems, metal or electronic implants, medications influencing neural excitability (antiepileptic, antipsychotics, or antidepressants), allergy to the electrode material, epilepsy, and pregnancy.

Sessions were conducted at the same time of day to minimise baseline EEG variances and subjects were allowed to take breaks in between experiment runs to prevent fatigue. Written informed consent was obtained from all participants. This study was approved by the Human Subjects Ethics Sub-committee of the Hong Kong Polytechnic University, and conducted according to the principles and guidelines of the Declaration of Helsinki.

### 2.2. Experimental Protocol

Participants underwent two EEG recording sessions on different days, based on a two-day crossover design. EEG and forearm EMG were measured as participants performed a movement task. Session A was performed with tSCS applied to the cervical region of the neck and Session B was performed without tSCS, on different days in order to minimise the potential of stimulation-induced brain activity changes. The order of sessions were randomised for each participant.

Participants undertook two activities during each session: A resting state task, and a movement execution task. During the former task participants were instructed to sit still for 90 s while their EEG was recorded. This was repeated twice: with eyes opened and eyes closed.

To assess the effect of tSCS on event-related desynchronisation of sensorimotor rhythms during movement, participants were instructed to perform rhythmic right-hand or bimanual finger flexion when cued by a computer screen. A rightwards arrow cued right-hand movement and a double arrow pointing both left and rightwards cued bimanual movements. Each movement was performed for four seconds and repeated 30 times, with a randomised 1.5 to 2.5 s inter-trial interval. EMG was recorded from the forearm muscles simultaneously to measure movement onset.

### 2.3. EEG/EMG Data Collection

EEG was recorded at 1200 Hz with a g.USBamp biosignal amplifier (g.tec, Schiedlberg, Austria). Nineteen passive electrodes were used: Fz, FC3, FC1, FCz, FC2, FC4, C3, C1, Cz, C2, C4, CP3, CP1, CPz, CP2, CP4, Pz, POz, and Oz, according to the international 10-20 system. The ground and reference electrodes were placed at AFz and right earlobe, respectively. EEG was internally filtered with a band-pass filter at 0.01–100 Hz, and notch filter at 50 Hz to attenuate powerline noise. Special attention was given to ensuring that electrode impedance was below 5 kΩ throughout the recording session, and that participants minimised their body movements. This was important as conventional data-cleaning techniques were made challenging by the presence tSCS. For consistency, typical rejection thresholds on peak-to-peak amplitudes were not used when processing either tSCS-off or tSCS-on data. EEG was pre-processed by applying a 3rd order Butterworth band-pass filter with a cutoff frequencies of 3 and 50 Hz.

Surface electromyography (EMG) was recorded from the left and right forearms to determine the beginning of movement onset and was used only in the movement classification analysis. Two electrodes (Ag/AgCl; F-301, Skintact, Innsbruck, Austria) were positioned on the belly of each extensor carpi radialis (ERC) muscle, with a 20 mm inter-electrode distance. Ground electrodes were attached to the lateral epicondyles. EMG was recorded simultaneously with EEG using a g.USBamp biosignal amplifier (g.tec, Schiedlberg, Austria) (Bandpass filter: 5–1200 Hz; notch filter: 50 Hz). Offline, a 20–500 Hz band-pass filter, and a 10 Hz low-pass filter were applied. Movement onset was defined as the moment the EMG signal exceeded the mean of the resting phase plus two times its standard deviation for at least 100 ms. EEG was epoched from −2 to 6 s relative to movement onset.

### 2.4. Transcutaneous Spinal Cord Stimulation (tSCS)

Stimulation was delivered in trains of ten 100 μs long biphasic rectangular pulses at a frequency of 30 Hz with a DS8R Biphasic Constant Current Stimulator (Digitimer, Hertfordshire, UK). The cathode electrode (3.2 cm diameter; Axelgaard Manufacturing Co, Fallbrook, CA, USA) was positioned between the C5-C6 intervertebral space. Hypoallergenic tape fastened the cathode to the skin to ensure snug contact throughout the session. Inter-connected anode electrodes (8.9 × 5.0 cm; Axelgaard Manufacturing Co, Fallbrook, CA, USA) were placed symmetrically on the shoulders, above the acromion. The current was determined as the highest intensity tolerable to the participant (40 ± 10 mA).

### 2.5. Artifact Suppression

To remove noise generated by tSCS we explored a number of artifact suppression techniques that could be implemented in real-time applications.

#### 2.5.1. Superposition of Moving Averages (SMA)

First developed to attenuate the effects of transcranial alternating current stimulation in EEG the SMA filter creates a template approximating the stimulation artifact and subtracts it from the contaminated EEG [12]. With each channel split into *N* non-overlapping windows of a length equal to the stimulation frequency the SMA filter averages *M* windows and subtracts the result from the current window, *n*. Hence, if x(n) is a single channel split into *N* segments,
(1)$$X\left(n\right)=x\left(n\right)-\frac{1}{M+1}\sum_{n-\frac{M}{2}}^{n+\frac{M}{2}}x\left(n\right),$$
where X(n) represents the cleaned, or ‘reconstructed’, channel. The artifact template is updated as it slides across the time series, adapting to changes in artifact shape.

In this study, M was set to 5, which was heuristically found to maximise the classification accuracy as explained later. This analysis was performed with code from an open source repository [13].

#### 2.5.2. Adaptive Filter

We also explored an adaptive filtering technique. Unlike conventional filters with fixed coefficients the adaptive filter adjusts its filtering parameters over time to satisfy an optimization algorithm. Many adaptive filters rely on two inputs, the corrupted signal and a signal reflecting known noise, often the output of the stimulator itself. We, however, implemented a version of the adaptive filter that relies only on the corrupted signal. A similar technique was used to remove functional electrical stimulation (FES) artifacts from EMG [14,15]. The method divides the incoming signal x(n) into *M* non-overlapping windows of *N* samples and makes a prediction of the stimulation artifact by using a linear combination of the *M* previous frames, weighted by filter coefficients *b*. It is assumed that if the filter can remove true EEG then the energy of the resulting signal will have a minimal value. Coefficient *b*, therefore, is determined by a least-squares algorithm which minimises the energy of the current frame with respect to this coefficient. A detailed explanation of this procedure was described by Sennels et al. [14]. Next, the predicted artifact is subtracted from the current frame:(2)$$y\left(n\right)=x\left(n\right)-\sum_{j=1}^{M}b_{j}x\left(n-jN\right),$$
where *N* is the ratio of the stimulation frequency to the sampling rate, ensuring that the stimulation artifact is aligned in each window. The subtraction of the predicted artifact from the current frame, x(n), aims to remove contributions from the stimulator, leaving behind a cleaned version of the signal, y(n).

This study found that *M* of 6 was generally enough to eliminate the simulation artifact.

#### 2.5.3. Median Filter

Neuromuscular electrical stimulation (NMES) has been delivered to peripheral musculature and shown to manifest in EEG as short latency, high amplitude peaks. Insautsti-Delago et al. applied a short sliding window to each EEG channel while taking the median value to curtail the effects of NMES [7]. The current study applied a similar method with a sliding window of 7 samples, or around 6 ms long.

#### 2.5.4. Notch Filter

A 3rd order Butterworth filter was used to attenuate the stimulation frequency by setting the low and high cut-off frequencies to 29 and 31 Hz, respectively.

### 2.6. Stimulation Artifact in the Time Domain

To illustrate the effect of tSCS on EEG in the time domain we plotted the pre-processed resting state EEG with eyes closed. In order to explore stimulation-intensity effects, we showed the EEG of the participants who were the most and least tolerant to tSCS. Furthermore, we investigated the impact of distance on artifact prominence by presenting data from the nearest and farthest channel to the stimulation site: Fz, and Oz, respectively. EEG from the tSCS-off condition is also shown for a better comparison.

### 2.7. Stimulation Artifact in the Frequency Domain

As tSCS is delivered at a fixed location on the posterior side of the neck we expected artifacts to manifest in EEG as a function of distance. For simplicity, we considered only the midline electrodes as we did not expect a lateralised effect owing to the relative homogeneity of scalp composition. To observe the effect we considered the power spectral density (PSD) of resting state EEG with eyes closed at and around the stimulation frequency (28–32 Hz). We expected the posterior electrodes (Oz, etc.) to have a greater 30 Hz contribution than the frontal electrodes (Fz, etc.). PSD was estimated using the multitaper method with a bandwidth of 0.1.

### 2.8. Spatial Distribution of tSCS Contamination

Using the method outlined above, we found the PSD of resting state EEG with eyes closed at and around 30 Hz to determine the spectral pattern of stimulation on scalp topography. The average power of each channel during tSCS was subtracted from and divided by the power from the tSCS-off condition, revealing the percentage power increase or decrease at the stimulation frequency. The process was repeated for the filtered EEG. Statistical differences in 30 Hz power between the tSCS-off and tSCS-on condition, and its filtered derivatives, were determined with a pairwise *t*-test where the data were found to follow a parametric distribution and a Wilcoxon signed-rank test where data were non-parametrically distributed. The *p*-values were adjusted using the Benjamini/Hochberg false discovery rate correction method.

### 2.9. Time Domain: EEG Descriptive Statistics

To characterise the EEG signal quantitatively and assess the impact of tSCS and artifact-suppression techniques we used a number of descriptive statistics. Namely, kurtosis, root-mean-square (RMS), Higuchi fractal dimension, and zero-crossings. This approach was motivated by a method proposed by Kohli et al. to evaluate the effectiveness of removing transcranial alternating current stimulation artifacts from EEG [12]. Eyes open, resting state EEG was used for this analysis. The EEG from each channel was split into 10 s non-overlapping segments. The descriptive statistics were calculated for each segment and averaged. An average was taken again across all participants and was displayed graphically.

The descriptive statistics across each EEG condition and electrode position were compared for significant differences. Firstly, the Levene and Shapiro–Wilk tests were performed to determine the homogeneity of variance and normality of the data. Where these tests were satisfied a two-way analysis of variance (ANOVA) was used with the descriptive statistic as the dependent variable and EEG condition and electrode position as the independent factors. The Scheirer-Ray-Hare test was performed where statistical distribution assumptions were not satisfied. Post-hoc tests for multiple comparisons included the pairwise *t*-test for descriptive statistics following a parametric distribution, and the Wilcoxon signed-rank test for non-parametric statistics. The *p*-values were adjusted using the Benjamini/Hochberg false discovery rate correction method.

### 2.10. Frequency Domain: Effect on Individual Alpha Frequency

The alpha rhythm is a prominent EEG feature which has been attributed to many cognitive processes [16]. To assess the feasibility of monitoring alpha rhythm expression during tSCS we extracted the peak frequency from the range of 8–12 Hz during the resting state task with eyes open and eyes closed. The PSD was calculated as outlined above. Normality and homogeneity were determined with the Shapiro–Wilk and Levene test, respectively. Where distribution assumptions were met we performed a one-way ANOVAs to determine if individual alpha frequency was significantly affected by EEG condition (tSCS-off, tSCS-on, tSCS-on with filters). We carried out multiple one-way ANOVAs for electrode location (Fz, Oz) and resting state (eyes open, eyes closed) given the strong differences in individual alpha frequency expected from the normal EEG. Post-hoc tests for multiple comparisons were performed as outlined above.

### 2.11. Classification of Sensorimotor Rhythms

To determine the feasibility of classifying movements from sensorimotor rhythms during tSCS we used the current state-of-the-art: Band-pass and common spatial pattern (CSP) filtering, followed by band-power feature extraction and linear discriminant analysis (LDA) classification [17]. EEG was divided into 2 second segments, 0.5–2.5 s, relative to movement onset. Two conditions were considered for classification: Right hand versus bimanual rhythmic finger flexion. Thirty trials per condition were used for training and testing the CSP-LDA classifier. The CSP approach consisted of finding spatial filters *w* such that the variance of the filtered EEG signals were maximal for one class and minimal for the other. Spatial filters *w* were found by extremising the following expression through a generalised eigenvalue decomposition:(3)$$\frac{wX_{1}X_{1}^{T}w^{T}}{wX_{2}X_{2}^{T}w^{T}},$$
where *T* denotes the transpose, and $X_{i}$ is multi-channel, bandpass filtered EEG from class *i*. Filter *w* contains a number of eigenvectors (spatial filters) corresponding to the number of EEG channels. It is best practice, however, to select several eigenvectors from each end of the eigenvalue spectrum as spatial filters to aid classification. In this study, we used six pairs of filters. Next, the logarithmic variance of the CSP-filtered EEG signals was used as features to train a LDA classifier. We used 10-fold cross-validation to evaluate the performance of the trained classifier. The accuracy of the classifier was defined as the number of correctly classified trials compared to the total number of trials. A pairwise *t*-test was used to determine statistically significant differences between the mean accuracies. The textitp-values were adjusted using the Benjamini/Hochberg false discovery rate correction method.

## 3. Results

### 3.1. Stimulation Artifact in the Time Domain

The time domain effects of stimulation intensity and electrode position on EEG are illustrated in Figure 1. Figure 1A,E show representative segments of eyes closed, resting state EEG from the participant whose EEG was least affected by tSCS, owing to them receiving only 10 mA of stimulation. The solid black line represents EEG recorded without tSCS and the grey dashed line is with tSCS. Visually, the signals in Figure 1A have a similar amplitude and both feature a 8–10 Hz component, typical of resting state EEG with eyes closed. It appears that at this intensity the frontal EEG channels are spared visually observable distortions. On the other hand, the posterior electrodes, represented by channel Oz (Figure 1E), show a clear 30 Hz component. The peak-to-peak amplitude at Oz is 120 μ V during tSCS compared with 30 μ V without tSCS, a 4-times increase.

At the other end of the intensity spectrum, Figure 1C,G show one second of resting state EEG from the participant who received the highest current intensity, 60 mA. In both Fz and Oz the EEG time series includes a substantial 30 Hz stimulation artifact, characterised as narrow high-amplitude peaks. It is most clearly visible in channel Oz. At 60 mA the stimulation condition increased the peak-to-peak amplitude 8.6-times, from 30 μV to 260 μV. The amplitude of the stimulation artifact at Fz is less intense, at around 4-times the size of normal EEG.

### 3.2. Stimulation Artifact in the Frequency Domain

Figure 1B,F show the power spectral density (PSD) of resting state EEG for a participant who received 10 mA of tSCS. Unlike in the time domain, where the presence of a stimulation artifact is unclear at channel Fz, Figure 1B displays a prominent peak at 30 Hz, and is even more pronounced at channel Oz, Figure 1F. This trend is mirrored in Figure 1D,H. The power is far greater in both channels, reflecting a much stronger current (60 mA). Outside of the 30 Hz frequency bin, the EEG spectra appear unaffected by tSCS compared to the PSD when tSCS is off.

### 3.3. Aliasing Effect

The tSCS artifact is not sufficiently captured by the EEG system, resulting in a constantly modulating artifact amplitude in the time domain (Figure 2A) and alternating power in the frequency domain (Figure 2B). The aliasing effect has been reported for other non-oscillatory, periodic stimulation techniques, for instance deep brain stimulation [10].

### 3.4. Spatial Distribution of tSCS Contamination

To determine how tSCS artifacts manifested in multi-channel EEG with respect to distance from the stimulation site we showed the normalized spectral power at the stimulation frequency (30 Hz) across the midline electrodes (Figure 3C). The topographic distribution of 30 Hz power relative to tSCS-off is also given in Figure 3A. Further, we explored whether this artifactual component could be removed in the frequency domain to the extent that it was statistically indistinguishable from tSCS-off. A Shapiro–Wilk test found that the power values tended to follow a non-parametric distribution. The following pairwise comparisons, therefore, used the Wilcoxon signed-rank test to assess statistically significant differences between power distributions.

It is evident from Figure 3B that the power at 30 Hz is substantially increased by tSCS in all electrodes with a rising intensity as a function of distance to stimulation site. Figure 3A illustrates this power increase when tSCS is present across the entire head. Compared to when no stimulation is applied the power at 30 Hz is increased by 900% at the posterior channels, with a gradual reduction in power moving from the occipital region but never returning to tSCS-off levels.

Figure 3C shows the power at 30 Hz once artifact-reduction techniques were applied. The 30 Hz power in each of the filtered signals was significantly reduced and better resembled the power of the tSCS-off condition, represented by the blue line. The distribution of 30 Hz power when tSCS is off tended to decrease from channel Fz to Pz, before increasing from Pz to Oz. Two filters were able to reproduce this distribution: the SMA (green line) and adaptive filter (red line).

The SMA filter removed the spectral pattern seen before artifact suppression, leaving a more evenly distributed power topography. Power at 30 Hz is diminished in all channels with a maximum difference of −40%.

The adaptive filter (A) diminished the stimulation artifact significantly but was still elevated compared to tSCS-on alone, the power is greatly diminished at only 58% above the tSCS-off session. Interestingly, the adaptive filter performed better on the posterior electrodes, which trended towards 0% modulation compared with no stimulation and with no statistically significant difference in means (*p* > 0.05). This perhaps suggests the adaptive filter is more effective where the stimulation artifact has a stronger signal-to-noise ratio.

The median filter resulted in the greatest underestimation of 30 Hz power in all channels. The notch filter (N), on the other hand, performed the best among the filters, suppressing the 30 Hz artifact, with statistically similar power at all midline electrodes (*p* > 0.05), except for Poz (*p* < 0.05) and Oz (*p* < 0.01).

### 3.5. Time Domain: EEG Descriptive Statistics

To quantify the EEG signals in the time domain we used descriptive statistics, see Figure 4. The Levene test and Shapiro–Wilk test showed that each descriptive statistic failed to meet assumptions of homogeneity of variance (*p* < 0.01) and normality (*p* < 0.01), respectively. The Scheirer-Ray-Hare test (similar to a two-way ANOVA but for non-parametric data) was therefore used to determine statistically significant effects based on condition, electrode, and condition-electrode interaction.

All descriptive statistics showed statistically significant differences based on condition (Kurtosis: *p* < 0.01; RMS: *p* < 0.01; Higuchi fractal dimension: *p* < 0.01; Zero crossings: *p* < 0.01) and electrode (Kurtosis: *p* < 0.01; RMS: *p* < 0.01; Higuchi fractal dimension: *p* < 0.01; Zero crossings: *p* < 0.01) but no interaction between condition and electrode (Kurtosis: *p* < 0.31; RMS: *p* < 0.90; Higuchi fractal dimension: *p* < 0.99; Zero crossings: *p* < 0.98). The results from a Wilcoxon signed-rank test for electrodes Fz and Oz are presented in Table 1.

A Wilcoxon signed-rank test showed that kurtosis was significantly different at CPz, Pz, POz, and Oz when tSCS was turned on. The SMA filter managed to transform the kurtosis at these channels to ranges statistically similar to that of EEG with tSCS turned off. The adaptive filter also performed well at POz and Oz but resulted in poorer signal reconstruction with significantly different kurtosis values (*p* < 0.01) at Fz, FCz, Cz, and Pz, compared to tSCS-off, perhaps implying the adaptive filter performs better on signals with well-defined artifacts.

The RMS was significantly elevated in all channels but more so at the posterior electrodes: From 5.87 to 8.37 μV at Fz (*p* < 0.01) and from 5.29 to 25.40 μV at Oz (*p* < 0.001). There were no filters which managed to suppress the tSCS contribution at all electrodes. The notch filter, however, performed the best at returning the RMS to levels statistically similar to that of clean EEG in five out of the seven midline electrodes investigated.

The Higuchi fractal dimension, a measure of signal complexity, was significantly altered at all channels once tSCS was applied (*p* < 0.001). The only filter able to suppress the tSCS-induced increase in complexity was the SMA filer which resulted in statistically similar values (*p* > 0.05) in all channels. The adaptive filter also performed well on POz and Oz. Interestingly, the notch filter increased the fractal dimension in all channels to an extent even greater than tSCS alone.

The number of zero crossings, a statistic that partly reflects signal frequency, was also significantly altered in all channels by tSCS. On average, the number of zero crossings per 10 s increased significantly from 22.12 to 29.0 (*p* < 0.001) at channel Fz and from 26.55 to 41.90 (*p* < 0.001) at channel Oz. Again, the SMA filter alleviated the effects of tSCS in all channels with 20.84 crossings per 10 seconds at Fz (a non-significant difference of −1.28, *p* = 0.077) and 23.60 at Oz (a non-significant difference of −1.16, *p* = 0.96). No other filter returned the average zero-crossings to levels statistically similar to that of tSCS-off EEG. The median filter significantly underestimated(*p* < 0.001), and the adaptive and notch filter significantly overestimated the number of zero crossings per 10 s (*p* < 0.001).

### 3.6. Frequency Domain: Individual Alpha Frequency

To assess how tSCS affected spectral features beyond the stimulation frequency we considered the individual alpha frequency for each participant at channel Fz and Oz, as illustrated in Figure 5. It is clear that the characteristic increase in peak alpha from the eyes open to eyes closed condition is displayed whether tSCS is applied or not. The normality and homogeneity were confirmed by the Shapiro–Wilk and Levene test, respectively. One-way ANOVAs were performed to compare the effect of EEG condition (tSCS-off, tSCS-on, tSCS-on with filters) on individual alpha peak frequency at during different resting states and channels. The analysis revealed that there was no statistically significant difference in individual alpha at Fz or Oz with eyes open or closed, see Table 2. A pairwise *t*-test for multiple comparisons found that the mean individual alpha peak frequency was not significantly different between any condition (*p* > 0.05), however at channel Fz the adaptive filtered EEG during eyes open neared a significant difference (*p* = 0.06).

### 3.7. Classification of Sensorimotor Rhythms

To determine the feasibility of classifying sensorimotor rhythms during tSCS we used EEG from a movement execution task to form a two-class classification problem. The 10-fold cross-validation scores are given in Table 3. The CSP-LDA method was able to predict correctly right-hand and bimanual finger flexion on average 76.14 ± 12.42% of the time when stimulation was off, and 75.71 ± 10.62 when stimulation was on. Both scores lie above chance level for a two-class BCI with 30 trials per class (67%, *p* < 0.05 [18]). A paired *t*-test reveled no statistically significant differences between these scores (*p* > 0.05). The *p*-values were adjusted using the Benjamini/Hochberg false discovery rate correction method.

Filtered EEG performed similarly well: SMA filter, 76.79 ± 9.51%; notch filter: 77.29 ± 11.17%; median filter: 77.14 ± 10.22%. Interestingly, these scores exceed the accuracies obtained using the tSCS-off and tSCS-on conditions, however not significantly so (*p* > 0.05). The adaptive filter performed poorly with 53.64 ± 12.24% accuracy, below chance level and therefore unsuitable for BCI applications.

## 4. Discussion

This study, for the first time, characterised the artifacts associated with transcutaneous electrical spinal cord stimulation in electroencephalography recordings. We found that tSCS produced narrow, high-amplitude peaks in the time domain at a rate equal to the stimulation frequency at nearly an order of magnitude more powerful than normal EEG. Through volume conduction, all electrodes were affected by tSCS to a greater or lesser extent. The degree of contamination was highly dependent on stimulation intensity and electrode position. We also found, however, that it may be possible to utilise EEG during tSCS, after applying artifact-suppression techniques. This study supports the use of a superposition of moving averages (SMA) filter as it resulted in descriptive statistics most resembling that of normal EEG. Kohli et al. drew similar conclusions in their report of removing electrical artifacts from EEG, which used a similar filter-evaluation strategy [12]. Perfect EEG reconstruction was not achieved with the SMA filter, however. Even after filtering the power at the stimulation frequency was different to normal EEG. It is a matter of contention whether perfect reconstruction is necessary in order to conduct legitimate analyses.

Analyses that do not overlap with the stimulation frequency may not require artifact-suppression processing at all. For instance, we showed that individual alpha frequencies can be extracted accurately during tSCS from all EEG channels, even without filtering. If an analysis must overlap with the stimulation frequency then a notch filter may be sufficient to reduce tSCS contamination to levels statistically similar as normal EEG, at least in the frontal and central electrodes. At the occipital area we showed that the adaptive filter was most effective in attenuating tSCS artifacts. The notch filter, however, may be too much of a blunt tool as it was unable to reconstruct the spatial distribution of spectral power typically associated with EEG [19]. The adaptive method, however, performed better at reconstructing the higher spectral power associated with the posterior channels, perhaps as the stimulation artifact is better defined and was therefore easier for the algorithm to remove.

Interestingly, the results from our movement-classification analysis found that spinal stimulation posed no impediment to BCI performance. SMA, notch, and median filtering actually increased the classification performance, but not significantly so, perhaps suggesting a potential neuromodulatory effect of tSCS. Indeed, high-intensity functional electrical stimulation has been shown to result in stronger event-related desynchronisation in the beta band (14–30 Hz) when applied to peripheral musculature, with an enhanced effect as a function of time [7,20]. The movement execution task in this study, however, involved only 30 repetitions of each movement. Future work should investigate whether this increase in performance would trend towards significance if stimulation were applied for a longer duration. The adaptive filter, however, should not be implemented in future analyses given its poor performance in this study, consistently yielding scores below chance level for a two-class BCI. As shown in the descriptive statistics analysis outlined above the adaptive filter performed better where the stimulation artifact is particularly prominent; that is, on EEG from the posterior electrodes. Given that most discriminatory motor signals come from the central area and the effects of stimulation are less prominent among these channels, the adaptive filter is likely poorly approximating the stimulation artifact and is removing valuable sensorimotor information instead. Although this filter has demonstrated efficacy in other work, these studies involved the reconstruction of EMG signals [14,15]. Therefore, it is likely not suitable for preserving the low-amplitude, low-frequency sensorimotor signals from EEG. Nevertheless, even without artifact-suppression, the tSCS-contaminated EEG proved classifiable with standard BCI techniques. This is a somewhat surprising result given the aliasing seen at the stimulation frequency. BCIs are often built around linear classifiers that require quasi-stationary band-power features to predict brain states. If the EEG power spectrum is exogenously modulated then band-power features likely carry less discriminatory power. Perhaps aliasing was not prominent enough given our 1200 Hz sampling rate to impact BCI performance. Future studies should bare this effect in mind, however, as a lower sampling rate would likely result in enhanced aliasing. Future studies should consider oversampling where practical [10].

Another practical consideration when performing an analysis on EEG recorded during tSCS is that it makes some conventional pre-processing steps challenging. For instance, many EEG pre-processing pipelines rely on rejection thresholds based on descriptive statistics—for example, channel amplitude, kurtosis, root-mean-square—to automatically remove bad spans of data [21,22]. As we have demonstrated here, EEG descriptive statistics are substantially altered by tSCS, meaning thresholding techniques would eliminate spans of data that are otherwise good. This may be a reason in itself to apply artifact-suppression techniques as a primary step in a tSCS analysis pipeline, particularly when working on EEG from posterior locations.

A potential limitation of this study is that the average stimulation intensity applied to the healthy volunteers (10–60 mA) was likely lower than what would be delivered in clinical practice. Spinal-cord injured individuals tend to have impaired sensibility below their injury level and can likely tolerate higher currents on average (40–200 mA). The results from this study therefore may not be representative of what is feasible in practice. Future analyses should replicate this study using a spinal-cord injured population to confirm if EEG monitoring is feasible at higher stimulation intensities.

The results from this study should not be viewed as a definitive statement on the effects of tSCS on EEG. Due to the variation in stimulation parameters used across tSCS studies conclusions can only be inferred with regards to the parameters that we have used here. For instance, we chose a one-millisecond long pulse with a 10 kHz carrier frequency delivered at 30 Hz to the cervical region of the spine, reflecting recent studies of upper-limb motor rehabilitation [2,3,23,24]. Other studies targeting lower limb rehabilitation or spasticity reduction have used different parameters: 20 or 50 Hz pulse trains, 5 kHz carrier frequencies, monophasic instead of biphasic pulses [25,26,27].

## 5. Conclusions

Owing to the relatively recent rise of tSCS there are many avenues of investigation currently unexplored. We note that investigations of cortical modulation have already begun and are likely to continue [6]. EEG offers invaluable access to brain dynamics, allowing source localisation and separation at excellent temporal resolutions [28]. This study provides an insight into the effects of cervical tSCS on EEG and our analyses showed that signal processing techniques such as the superposition of moving averages filter can reasonably suppress tSCS contamination. We conclude that simultaneous EEG monitoring is feasible and reliable, and encourage subsequent research to use EEG to better understand the activity of the sensorimotor cortex during tSCS-based rehabilitation of spinal-cord injury patients.

## Figures and Tables

**Figure 1 sensors-21-06593-f001:**
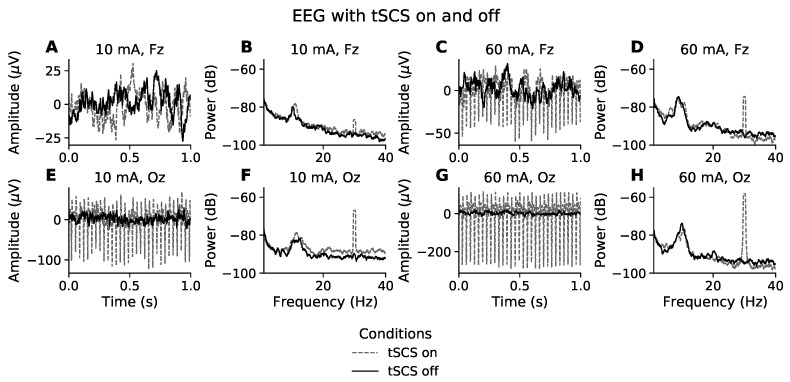
Effect of tSCS on EEG in the time and frequency domain. Time domain (**A**,**C**,**E**,**G**): The first and third column show one second of resting state, eyes closed EEG for the participants with the lowest (10 mA) and highest (60 mA) tolerance to stimulation intensity, respectively. The EEG channel farthest from the stimulation site (Fz) is represented in the first row while the second row relates to the channel most proximal to the stimulation site (Oz). Frequency domain (**B**,**D**,**F**,**H**): The second and fourth column show the power spectral density of resting state, eyes closed EEG for the participants with the lowest (10 mA) and highest (60 mA) tolerance to stimulation intensity, respectively. EEG with stimulation on and stimulation off are presented with grey dashed and solid black lines, respectively.

**Figure 2 sensors-21-06593-f002:**
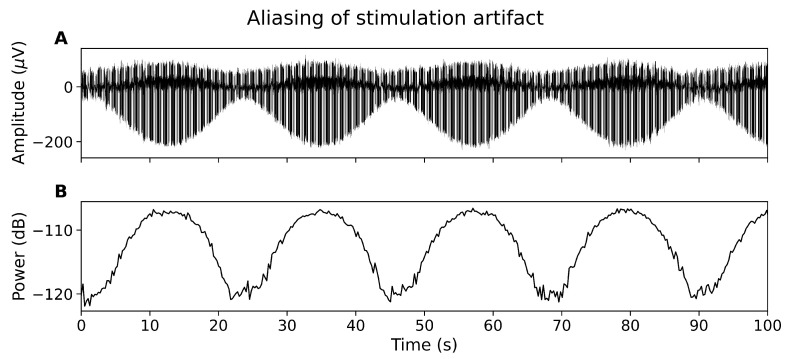
The aliasing effect. (**A**) EEG showing amplitude of tSCS peaks changing over time. (**B**) Power spectral density at 30 Hz over time.

**Figure 3 sensors-21-06593-f003:**
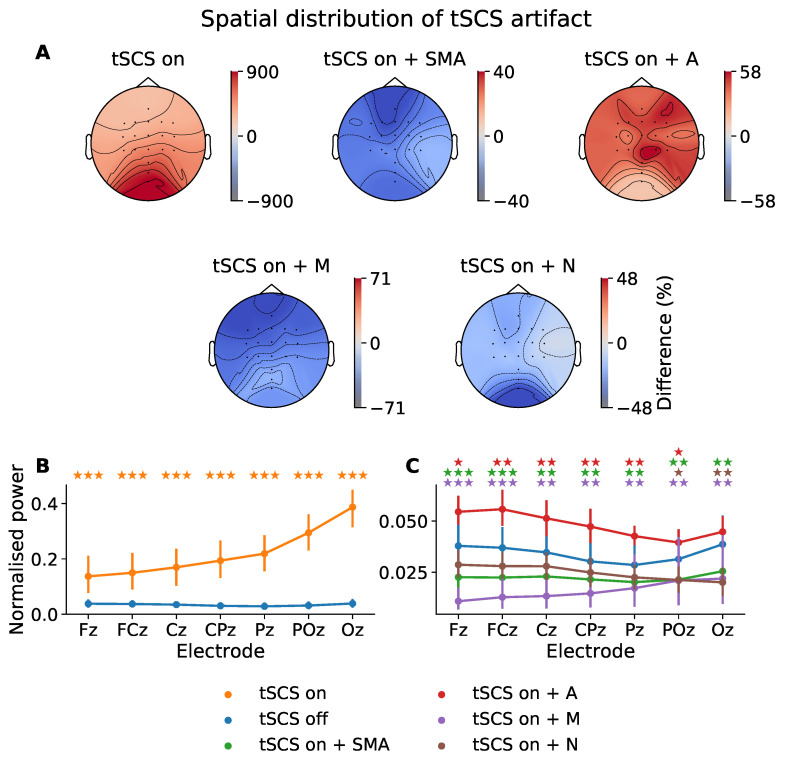
Power distribution at 30 Hz (i.e., the stimulation frequency) across all participants. (**A**) Topographic power differences of tSCS-on, and its filtered derivatives, relative to tSCS-off (%). SMA: superposition of moving averages filter; A: adaptive filter; M: Median filter; N: notch filter. (**B**) Normalised spectral power at 30 Hz across midline electrodes during tSCS-on and tSCS-off. (**C**) Normalised spectral power at 30 Hz across midline electrodes for tSCS-off and tSCS-on after artifact-suppression. The *p*-values from a Wilcoxon signed-rank test between tSCS-off and each tSCS-on condition are indicated with a colour-coded star for each electrode (★ *p* < 0.05, ★★*p* < 0.01, ★★★
*p* < 0.001).

**Figure 4 sensors-21-06593-f004:**
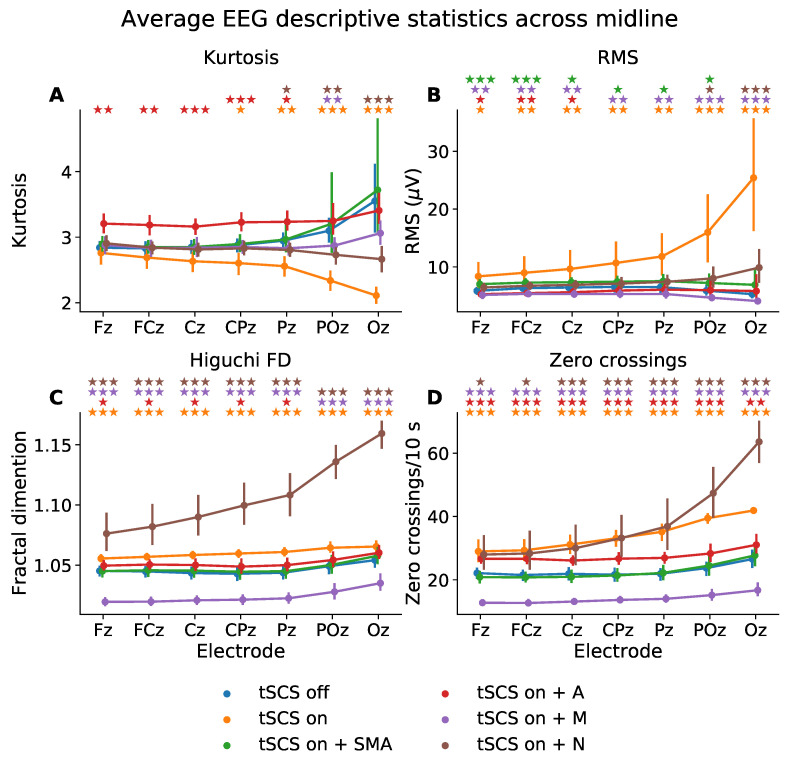
Mean descriptive statistics—(**A**) kurtosis, (**B**) root mean square (RMS), (**C**) Higuchi fractal dimension (FD), (**D**) zero-crossings—of eyes closed, resting state EEG from midline electrodes, for tSCS-off, tSCS-on, and tSCS-on with filtering. SMA: superposition of moving average filter; A: adaptive filter; M: median filter; N: notch filter. The *p*-values from a Wilcoxon signed-rank test between tSCS-off and each tSCS-on condition are indicated with a colour-coded star for each electrode (★ *p* < 0.05, ★★*p* < 0.01, ★★★
*p* < 0.001).

**Figure 5 sensors-21-06593-f005:**
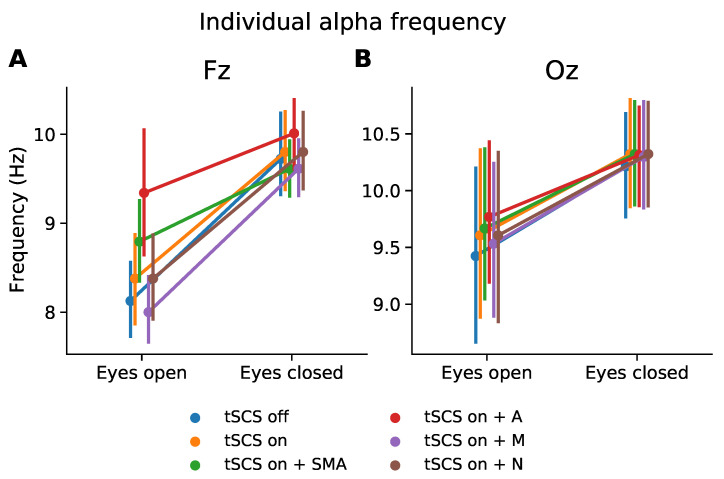
Subject-wise peak frequency in alpha range (8–12 Hz) during resting state with eyes opened and eyes closed. (**A**) Channel Fz, (**B**) Channel Oz.

**Table 1 sensors-21-06593-t001:** Participant-wise mean of EEG descriptive statistics—kurtosis, root mean square (RMS), Higuchi fractal dimension, zero-crossings—of resting state EEG with eyes open. The difference between each tSCS-on condition with respect to tSCS-off is given in addition to the *p*-values associated with a Wilcoxon signed-rank test as these descriptive statistics were found to be non-parametric. The *p*-values were adjusted using the Benjamini/Hochberg false discovery rate correction method.

	Fz			Oz		
	Kurtosis (Unitless)					
	Mean	*p*-Value	Difference	Mean	*p*-Value	Difference
tSCS-off	2.84	-	-	3.55	-	-
tSCS-on	2.76	0.61	−0.08	2.11	<0.001	−1.44
tSCS-on + adaptive	3.21	<0.01	0.37	3.41	0.97	−0.15
tSCS-on + median	2.88	0.70	0.04	3.06	0.33	−0.50
tSCS-on + notch	2.90	0.47	0.07	2.66	<0.001	−0.89
tSCS-on + SMA	2.90	0.49	−0.06	3.72	0.69	0.17
	RMS (μ V)					
	Mean	*p*-value	Difference	Mean	*p*-value	Difference
tSCS-off	5.87	-	-	5.29	-	-
tSCS-on	8.37	0.01	2.50	25.40	<0.001	20.11
tSCS-on + adaptive	5.28	0.05	−0.58	5.83	0.13	0.54
tSCS-on + median	5.15	<0.01	−0.72	4.09	<0.01	−1.20
tSCS-on + notch	6.45	0.16	0.58	9.88	<0.01	4.60
tSCS-on + SMA	7.02	<0.001	1.15	6.9	0.11	1.61
	Higuchi fractal dimension					
	Mean	*p*-value	Difference	Mean	*p*-value	Difference
tSCS-off	1.045	-	-	1.054	-	-
tSCS-on	1.056	<0.001	0.01	1.065	<0.01	0.01
tSCS-on + adaptive	1.050	<0.01	0.004	1.060	<0.01	0.006
tSCS-on + median	1.020	<0.001	−0.03	1.035	<0.001	−0.02
tSCS-on + notch	1.086	<0.001	0.03	1.159	<0.001	0.11
tSCS-on + SMA	1.045	0.93	−0.0002	1.058	0.30	0.004
	Zero crossings (Crossings/10 s)					
	Mean	*p*-value	Difference	Mean	*p*-value	Difference
tSCS-off	22.12	-	-	26.55	-	-
tSCS-on	29.0	<0.001	6.85	41.90	<0.001	15.34
tSCS-on + adaptive	26.64	<0.001	4.52	31.05	<0.01	4.50
tSCS-on + median	12.76	<0.001	−9.36	16.70	<0.001	−9.85
tSCS-on + notch	27.97	<0.01	5.85	63.60	<0.001	37.05
tSCS-on + SMA	20.84	0.077	−1.28	27.71	0.96	1.16

**Table 2 sensors-21-06593-t002:** Subject-wise average of individual alpha peak frequencies. The result of a one-way ANOVA is given for the ‘eyes open’ and ‘eyes closed’ condition. A pairwise *t*-test for multiple comparisons determined if the mean of each condition was significantly different from the tSCS-off condition. The *p*-values were adjusted using the Benjamini/Hochberg false discovery rate method. The difference with the tSCS-off condition is given. Shapiro–Wilk’s and Levene’s tests were performed to confirm normality and homogeneity before each ANOVA (*p* < 0.05).

	Fz					
	Eyes open			Eyes closed		
	F(5,102) = 3.52, *p* = 0.069, $\eta^{2}$ = 0.15			F(5,102) = 0.50, *p* = 0.77, $\eta^{2}$ = 0.024		
	Mean	*p*-value	Difference	Mean	*p*-value	Difference
tSCS-off	8.13	-	-	9.72	-	-
tSCS-on	8.38	0.54	0.25	9.80	0.91	0.08
tSCS-on + adaptive	9.34	0.06	1.21	10.01	0.85	0.29
tSCS-on + median	8.00	0.72	−0.13	9.61	0.94	−0.11
tSCS-on + notch	8.38	0.54	0.25	9.80	0.94	0.08
tSCS-on + SMA	8.78	0.10	0.67	9.61	0.94	−0.11
	Oz					
	Eyes open			Eyes closed		
	F(5,102) = 3.52, *p* = 0.99, $\eta^{2}$ = 0.0050			F(5,102) = 0.031, *p* = 0.99, $\eta^{2}$ = 0.0015		
	Mean	*p*-value	Difference	Mean	*p*-value	Difference
tSCS-off	9.42	-	-	10.21	-	-
tSCS-on	9.61	0.97	0.18	10.32	1.0	0.11
tSCS-on + adaptive	9.77	0.97	0.34	10.31	1.0	0.1
tSCS-on + median	9.53	0.97	0.11	10.30	1.0	0.08
tSCS-on + notch	9.61	0.97	0.18	10.32	1.0	0.11
tSCS-on + SMA	9.67	0.97	0.24	10.32	1.0	0.1

**Table 3 sensors-21-06593-t003:** Mean 10-fold classification accuracies across all subjects. The significance level of a paired *t*-test is given with respect to the tSCS-off condition.

	Accuracy (%)	*p*-Value
tSCS-off	76.14 ± 12.42	-
tSCS-on	75.71 ± 10.62	0.84
tSCS-on + SMA	76.79 ± 9.51	0.76
tSCS-on + adaptive	53.64 ± 12.24	0.00015
tSCS-on + notch	77.29 ± 11.17	0.6
tSCS-on + median	77.14 ± 10.22	0.55

## Data Availability

Data will be made available upon reasonable request to the authors.

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
