# Peer review of "EEG Monitoring Is Feasible and Reliable during Simultaneous Transcutaneous Electrical Spinal Cord Stimulation"

_sensors, 2021, doi:10.3390/s21196593_

Round 1

Reviewer 1 Report

The paper addresses the feasibility of simultaneous transcutaneous electrical spinal cord stimulation (tSCS) and electroencephalography (EEG) acquisition. The principal aim is to describe the artifacts induced by tSCS in EEG recordings of healthy subjects and test possible artifacts-suppression techniques. The manuscript is well written and clearly describes the methodology followed. As stated by the authors themselves in the discussion section, the study presents some limitations due to experimental choices, e.g., only healthy subjects were evaluated, with a relatively low stimulation intensity. However, the results presented are a first step in the characterization of simultaneous tSCS-EEG application and it can be of interest for the community of researchers approaching the analysis of these signals. Therefore, I recommend its publication in Sensors.

I have a few minor suggestions to improve readability:

  1. In section Materials and Methods, it is declared that the resting state EEG recordings where performed both with open and closed eyes, however it is not clear in which of the two condition the analysis described in paragraphs 2.6, 2.7 and 2.8 were performed. In figure 1 caption, for example, only closed eyes signal is reported. Please specify this aspect.
  2. Please, comment the choice declared in paragraph 2.7 to observe the effect of stimulation artifact in spatial and frequency domain only in the midline electrodes and why a lateralized spatial effect was a priori excluded.
  3. Paragraph 2.11: the “LDA” acronym seems to appear before its definition “Linear Discriminant analysis” at line 225. Consider rearranging the paragraph structure, thus introducing the reader to the different steps of the classification method with order.
  4. In Figure 1, use a uniform y-axis scale among panels. This would allow a better comparison between the different stimulation effects, at least for Power Spectral Density.
  5. The time domain EEG descriptive statistics reported in paragraph 3.5 was performed only in the closed eyes condition. Was a comparison with the opened eyes EEG features considered? Since more artifactual components are usually expected with open eyes, it would be of interest reporting on the difference, if any, between the EEG characteristics in the two conditions and how this could affect the artifacts-correction techniques employed.
  6. Again in paragraph 3.5. Statistical comparison between electrodes are reported only for Fz and Oz, while in Figure 4 all the midline electrodes are shown. For completeness, all electrodes considered should be accounted in the statistical analysis.
  7. For a better comprehension of the results, I suggest to report in Figure 3 at least the statistical significant differences found between electrodes and/or techniques, for example with a star for significance.
  8. In figure 4, some of the colours used are not well distinguishable. The use of different symbols for different series could help.
  9. At line 339, it is not clear what it is meant for “more linear relationship”. Please quantify it, if possible.
  10. A final separate section would be nice to clearly draw conclusions of the work after the discussion.
  11. Check a few typos, as for example:

line 41: “corticospincal” should be “corticospinal”

line 225 “discriminate analysis …” should be “discriminant analysis…”

Figure 1 caption: “power spectrum density” should become “power spectral density”

Reviewer 2 Report

Authors have undertaken the project to characterise the high amplitude artifacts that are manifested in EEG when recorded simultaneously with tSCS (transcutaneous electrical spinal cord stimulation) application. The problem is essential, because tSCS is more and more frequently used in the treatment of patients after incomplete spinal cord injuries, and any neurophysiological method for monitoring the activity in supraspinal neuronal centers (both cortical and subcortical) is of great interest and useful for clinical evaluation of tSCS effectiveness. Moreover studies may provide the insight on the neural mechanisms „underpinning” (cit.). sensory and motor recovery following tSCS application.

I have only some minor more propositions than revisions:

Title

Because authors found hard evidences that …the superposition of moving averages filter was the most successful technique at returning contaminated EEG to levels statistically similar to that of normal EEG, and in the frequency domain, notch filtering was more effective at reducing the spectral power contribution of stimulation from frontal and central electrodes…, I will propose to enrich the title to „EEG is Feasible and Reliable monitoring method during Simultaneous Transcutaneous Electrical Spinal Cord Stimulation”

Abstract

The “able-bodied participants” introduces a suspicion to the reader, rarely used in medical papers, could it be replaced with “healthy volunteers” throughout the text?

The final conclusion is missing in the first part of the sentence, in the second it expresses the true: (cit.) „This study supports future investigations using EEG for studying the corticospinal mechanisms of sensorimotor recovery, and potentially paves the way to brain-computer interfaces operating in the presence of spinal stimulation.”

EEG does not study the “corticospinal mechanisms of sensorimotor recovery” in this paper considering the obtained results, but …motor cortex neural centers activity following tSCS application,…

Materials and Methods

Exclusion criteria should be supplemented with “cardiovascular diseases”, “musculoskeletal pathology” is too general – I will suggest “previous neurological symptoms in nervous and musculo-skeletal systems”

Other sections are perfectly presented. Congratulations.
